# CoDBench: A Critical Evaluation of Data-driven Models for Continuous Dynamical Systems

## Abstract

Continuous dynamical systems, characterized by differential equations, are ubiquitously used to model several important problems: plasma dynamics, flow through porous media, weather forecasting, and epidemic dynamics. Recently, a wide range of data-driven models has been used successfully to model these systems. However, in contrast to established fields like computer vision, limited studies are available analyzing the strengths and potential applications of different classes of these models that could steer decision-making in scientific machine learning. Here, we introduce CoDBench, an exhaustive benchmarking suite comprising 12 state-of-the-art data-driven models for solving differential equations. Specifically, we comprehensively evaluate 4 distinct categories of models, *viz.*, feed forward neural networks, deep operator regression models, frequency-based neural operators, and transformer architectures against 10 widely applicable benchmark datasets encompassing challenges from fluid and solid mechanics. We conduct extensive experiments, assessing the operators' capabilities in learning, zero-shot super-resolution, data efficiency, robustness to noise, and computational efficiency. Interestingly, our findings highlight that current operators struggle with the newer mechanics datasets, motivating the need for more robust neural operators. All the datasets and codes are shared in an easy-to-use fashion for the scientific community. We hope this resource will be an impetus for accelerated progress and exploration in modeling dynamical systems. For codes and datasets, see: https://anonymous.4open.science/r/cod-bench-7525.

## 1 Introduction

Nature is in a continuous state of evolution. "Rules" governing the time evolution of systems in nature, also known as dynamics, can be captured mathematically through partial differential equations (PDEs). In the realm of science and engineering, PDEs are widely used to model and study several challenging real-world systems, such as fluid flow, deformation of solids, plasma dynamics, robotics, mechanics, and weather forecasting, to name a few (Debnath & Debnath, 2005; Nakamura, 1977; Robert, 2007). Due to their highly non-linear and coupled nature, these PDEs can be solved analytically only for trivial or model systems. Thus, accurate numerical solutions for the PDEs are the cornerstone in advancing scientific discovery. Traditionally, PDEs are solved using classical numerical methods such as finite difference, finite volume, or finite element methods (Sewell, 2012). However, these numerical methods exhibit major challenges regarding system size, timescales, and numerical instabilities in realistic systems. Specifically, simulating the systems for longer timescales or large domains is extremely computationally intensive, so performing them in real-time for decision-making is a major challenge. Further, in the case of large/highly non-linear fields, these simulations often exhibit numerical instabilities, rendering them ineffective (Šolín, 2005).

The recent surge in artificial intelligence-based approaches suggests that neural models can efficiently capture continuous dynamical systems in a data-driven fashion (Brunton & Kutz, 2022). These models are more time-efficient than traditional solvers and can capture highly non-linear input-output relationships. Earlier approaches in this direction relied on learning the input-output map through multilayer perceptrons, convolutional neural networks, or graph neural networks. However, these approaches faced challenges in generalizing to unseen boundary conditions, geometries,

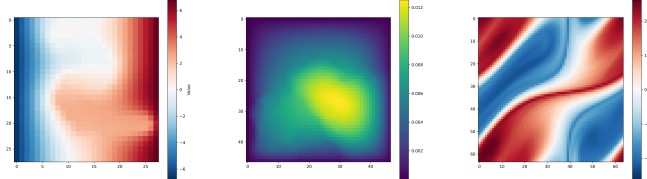

Figure 1: A glimpse of the data diversity under consideration. From left to right, we present visual representations of three distinct datasets — Biaxial, Darcy Flow, and Navier-Stokes.

or resolutions. This could be attributed to the fact that the neural models essentially learn the input-output relationship in a finite-dimensional approximation. To address this challenge, a seminal theory, extending the universal approximation theorem of neural networks (Cybenko, 1989) to neural operators was proposed, namely, the universal operator approximation theory (Chen & Chen, 1995). This unveiled the networks' prowess in handling infinite-dimensional inputs/ outputs.

Theoretically, directly learning the solution operator through specialized neural network architectures offers several key advantages. (i) They can directly learn input-output function mappings from data, thereby obviating the necessity for prior knowledge of the underlying PDE. (ii) They offer significantly improved time efficiency compared to traditional numerical solvers. (iii) They exhibit zero-shot generalizability, extending their applicability to systems of larger scale and complexity than those encompassed within the training dataset. (iv) They provide superior approximations of the solution operator compared to existing neural architectures, spanning from feed-forward networks to specialized models like CNNs' and GANs.' Thus, the neural operators attempt (Kovachki et al., 2021) to combine the best data-driven and physics-based numerical models.

This motivated the exploration of neural operator architectures (Bhattacharya et al., 2021), (Nelsen & Stuart, 2021), capable of directly learning the solution operator. For instance, consider DEEP-ONET (Lu et al., 2021a), which leverages the universal approximation theorem introduced by Chen and Chen to address PDEs directly. On a different front, FNO (Li et al., 2020), one of the most widely used Neural Operators, focuses on parameterizing the integral kernel within Fourier space. Moreover, a noteworthy study (Cao, 2021) highlights the notion that all transformers are essentially operators. This insight has sparked endeavors to create operator transformers. Given their proven effectiveness in sequence-to-sequence learning tasks, these transformer-based designs open avenues for enhancing the approximation of spatiotemporal PDEs. Prior studies, such as those by (Hao et al., 2022), have delved into the realm of PINNs (Raissi et al., 2019) and some neural operator architectures, like DEEPONET, FNO, and their variants. Unlike fields like computer vision, there's a lack of comprehensive comparative evaluations for these neural operators. The inherent variations and incompatibilities in architectures make direct comparisons extremely cumbersome. These evaluations are crucial for discerning the unique advantages of different architectural paradigms, particularly when addressing equations across diverse scientific domains. While several attempts at a codebase with multiple neural operators exist, current works include limited operators (see Table:9).

This study aims to bridge this gap by rigorously evaluating data-driven models encompassing various classes and methods, including the foundational deep operator regression model, frequency domain parameterization models, and transformer-based architectures, to achieve state-of-the-art performance comparisons on selected PDE datasets. Our dataset selection is methodical, designed to challenge each model with equations from various scientific disciplines, (see Fig.1 for a teaser). We incorporate five prevalent equations from fluid dynamics and five standard differential equations from solid mechanics into the neural operator domain, ensuring a holistic comparison within the realm of neural operators. In this work, we critically analyze 12 data-driven models, including operators and transformers, on 10 PDE datasets. Specifically, our contributions are as follows:

1. **CODBENCH:** We present a package that allows seamless analysis of several data-driven approaches on PDEs. We thoroughly assess state-of-the-art data-driven models for solving PDE datasets across diverse scientific realms, shedding light on their precision and efficacy.

2. **Super-resolution:** We analyze the ability of neural operators to generalize to systems of different resolutions than that of their training sets' discretizations.

3. **Data efficiency and robustness to noise:** We critically assess the efficiency of these models to learn from small amounts of data or noisy data. This is an important aspect since the data available can be scarce and noisy in practical applications.

4. **Out-of-distribution task:** A novel task to gain insights into what these models are truly learning to determine whether the underlying operator was genuinely being learned or the training dataset was overfitting. Two closely related STRESS and STRAIN datasets are interchanged during training and testing to investigate whether the solvers are operators.

## 2 PRELIMINARIES

This section provides a concise mathematical framework to illustrate how traditional PDE solving can be transitioned and addressed using data-driven methodologies via neural networks.

1. **Function Domains**: Consider a bounded open set, represented as $\mathcal{D} \subset \mathbb{R}^d$. Within this domain, we define $\mathcal{F} = \mathcal{F}(\mathcal{D}; \mathbb{R}^{d_f})$ and $\mathcal{G} = \mathcal{G}(\mathcal{D}; \mathbb{R}^{d_g})$ as separable Banach spaces, corresponding to input and output functions, which are elements in $\mathbb{R}^{d_f}$ and $\mathbb{R}^{d_g}$, respectively.

2. **The Solution Operator**: In our exploration, we introduce $T^\dagger : \mathcal{F} \to \mathcal{G}$, a mapping that is typically nonlinear. This mapping emerges as the solution operator for PDEs, playing a pivotal role in scientific computations.

3. **Data Generation**: For training purposes, models utilize PDE datasets constructed as $\mathcal{D} = \{(\mathcal{F}_k, \mathcal{G}_k)\}_{1 \leq k \leq D}$, where $\mathcal{G}_k = T^\dagger(\mathcal{F}_k)$. Given the inherent challenges in directly representing functions as inputs to neural networks, the functions are discretized using mesh generation algorithms (Tristano et al., 1998) over domain $\mathcal{D}$. We sample both input and output functions on a uniform grid, as it ensures compatibility with all selected solvers. For the input function $\mathcal{F}_k$, we discretize it on the mesh $\{x_i \in \Omega\}_{1 \leq i \leq R}$, and the discretized $\mathcal{F}_k$ is $\{(x_i, f_{ik})\}_{1 \leq i \leq R}$, where $f_{ik} = \mathcal{F}_k(x_i)$. Similarly, For the solution function $\mathcal{G}_k$, we discretize it on the mesh $\{y_i \in \Omega\}_{1 \leq i \leq R}$, and the discretized $\mathcal{G}_k$ is $\{(y_i, g_{ik})\}_{1 \leq i \leq R}$, where $g_{ik} = \mathcal{G}_k(y_i)$. It's worth noting that models such as POD-DEEPONET and SNO utilize only the function values for representation, excluding grid locations from the model input.

4. **Objective**: The overarching goal for each model is to craft an approximation of $T^\dagger$. This is achieved by developing a parametric mapping, denoted as $T : \mathcal{F} \times \Theta \to \mathcal{G}$ or, in an equivalent form, $T_\theta : \mathcal{F} \to \mathcal{G}$, where $\theta \in \Theta$, is a parameter space.

5. **Metric**: Evaluating the efficacy of the parametric mapping involves comparing its outputs, $T_\theta(\mathcal{F}_k) = \{\widetilde{g}_{ik}\}_{1 \leq i \leq R}$, with the actual data, aiming to minimize the relative L2 loss:

$$\min_{\theta \in \Theta} \frac{1}{D} \sum_{k=1}^{D} \frac{1}{R} \frac{\|T_\theta(\mathcal{F}_k) - \{\widetilde{g}_{ik}\}_{1 \leq i \leq R}\|_2^2}{\|\{\widetilde{g}_{ik}\}_{1 \leq i \leq R}\|_2^2}, \tag{1}$$

Here, $R$ denotes the function discretization parameter.

## 3 MODEL ARCHITECTURES

A selection of 12 data-driven models is made, spanning four different categories (see Fig 2). Standard neural network architectures are incorporated to establish a baseline for all neural operators, while deep operator regression models serve as foundational elements. Advanced state-of-the-art contributions emerge in frequency-based operators, with FNO representing a milestone model. Including frequency-based models enhances the benchmark's utility. Notably, 2023 introduces three major transformer-based neural operator architectures, exhibiting state-of-the-art performance in error rates. The inclusion of transformer-based models aims to compare the latest research in neural operators with previously established approaches.

**Standard Neural Network Architectures:** UNET, delineated in (Ronneberger et al., 2015), employs a U-shaped encoder-decoder design augmented by skip connections, facilitating the capture of granular and abstract features. RESNET, described in (Jian et al., 2016), consists of a series of residual blocks and are commonly used in computer vision tasks (Targ et al., 2016). Conditional Generative Adversarial Networks (CGAN), introduced in (Mirza & Osindero, 2014), are an evolution of the GAN framework, facilitating conditional generation via the incorporation of label information in both the generator and discriminators. FNN is a foundational element for all machine learning models. Meanwhile, RESNET and UNET exhibit promising results, with UNET demonstrating competitive error rates among leading neural operators. CGAN features a distinctive generative-adversarial architecture, showing promise in learning PDE datasets. Including these models in the benchmark study reflects their simple architecture yet notable performance in addressing partial differential equations.

**Deep Operator-Based Regression Models:**

Neural Operators represent a novel ML paradigm, predominantly employed in scientific machine learning to decipher PDEs. These operators rely solely on data and remain agnostic to the underlying PDE. DEEPONET bifurcates into two sub-networks: the branch net, which encodes the input function at fixed sensor locations, and the trunk net, encoding solution function locations (Lu et al., 2021a). The solution emerges from the inner product of the outputs from these nets. In POD-

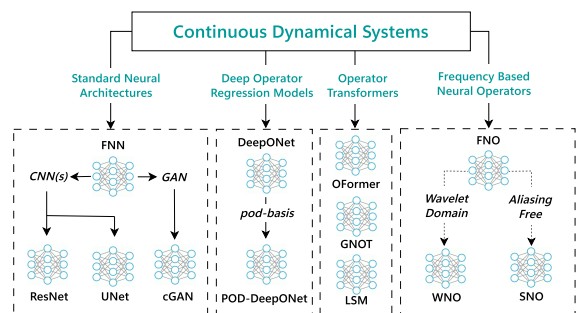

Figure 2: An overview of the various models being benchmarked and their relationship. The term 'pod-basis' denotes the basis of the output function, derived directly from proper orthogonal decomposition instead of being learned through a neural network.

DEEPONET, the bases are determined by executing proper orthogonal decomposition (POD) on training data, replacing the self-learned basis of output functions (Lu et al., 2022). This POD basis forms the trunk net, leaving only the branch net as the trainable component, which discerns the coefficients of the POD basis. These models exemplify a direct application of the universal operator approximation theory, highlighting the capacity of neural networks in learning and embodying operators.

**Frequency-Based Operators:** Frequency-based solvers like FNO employ a finite-dimensional parameterization using truncated Fourier modes (Li et al., 2020). By integrating this with an integral operator restricted to convolution and instantiated via a linear transformation in the Fourier domain, the FNO operator is conceived. WNO, or Wavelet Neural Operator, amalgamates the prowess of wavelets in time-frequency localization with an integral kernel. By learning the kernel in the wavelet domain, convolution operates on wavelet decomposition coefficients rather than direct physical space convolution (Tripura & Chakraborty, 2022). SNO, the Spectral Neural Operator, addresses the often-overlooked aliasing error in the Fourier Neural Operator. By representing both input and output functions using coefficients in truncated Fourier or Chebyshev series, SNO offers an aliasing-free approach (Fanaskov & Oseledets, 2022). Any transformation between these coefficients can be executed using neural networks, and methods employing these series are termed spectral neural operators. Their approach utilizes a straightforward, feed-forward neural network architecture in the complex domain. Although multiple variants of FNO demonstrate commendable performance, the selected models above are chosen to exemplify the most diverse yet effective approaches in solving partial differential equations (PDEs) within the frequency domain.

**Transformer Operators:** GNOT introduces the Heterogeneous Normalized (linear) Attention (HNA) block and a geometric gating mechanism, specifically tailored for enhanced performance on PDE datasets (Hao et al., 2023). This architecture effectively performs a soft domain decomposition (Jagtap & Karniadakis, 2021), treating each decomposed domain independently and subsequently integrating them using a mixture-of-experts approach to predict the underlying truth. In contrast, the OFORMER model builds upon the seminal work presented in (Vaswani et al., 2017). It incorporates random Fourier projection to counteract spectral bias, enhancing its efficacy on PDEs Li et al. (2022a). Another recent advance in the field of the neural operator comes with the Latent Spectral Model. It maps the high-dimensional coordinate space to a compact latent space and learns the solution operator in latent space. An attention-based neural network is instantiated to learn the mapping from coordinate to latent space and vice-versa (Wu et al., 2023). Despite sharing a common foundational architecture, all the aforementioned models approach the task of learning partial differential equations (PDEs) differently. Their varied methodologies and consistent state-of-the-art performance across multiple datasets make them ideal candidates for inclusion in the evaluation of data-driven PDE solvers.

## 4 DATASETS

Here, we briefly describe the 10 datasets used in the present. While previous approaches have mostly focussed on fluid datasets, here we present 5 datasets on fluid flow and 5 on the deformation of solids; for complete dataset details, refer A.1.

| Models | Datasets | | | | | | | |
|---|---|---|---|---|---|---|---|---|
| | **BURGERS** | **DARCY** | **NAVIER STOKES** | **SHALLOW WATER** | **STRESS** | **STRAIN** | **SHEAR** | **BIAXIAL** |
| FNN | $5.853_{\pm 1.416}$ | $3.47_{\pm 0.14}$ | $34.77_{\pm 0.19}$ | $2.424_{\pm 0.656}$ | $25.69_{\pm 0.59}$ | $23.09_{\pm 7.08}$ | $1.11_{\pm 0.06}$ | $3.69_{\pm 0.01}$ |
| RESNET | $11.327_{\pm 1.208}$ | $5.14_{\pm 0.23}$ | $29.52_{\pm 0.14}$ | $0.287_{\pm 0.093}$ | $20.05_{\pm 0.19}$ | $14.64_{\pm 0.31}$ | $3.02_{\pm 0.95}$ | $13.58_{\pm 2.67}$ |
| UNET | $30.870_{\pm 2.000}$ | $2.10_{\pm 0.08}$ | $24.02_{\pm 0.95}$ | $0.295_{\pm 0.097}$ | $10.57_{\pm 0.19}$ | $9.05_{\pm 0.33}$ | $7.09_{\pm 0.46}$ | $16.63_{\pm 2.30}$ |
| cGAN | $34.906_{\pm 0.506}$ | $1.88_{\pm 0.04}$ | $24.00_{\pm 0.48}$ | $0.291_{\pm 0.027}$ | $6.66_{\pm 0.84}$ | $6.12_{\pm 0.80}$ | $5.63_{\pm 0.50}$ | $15.74_{\pm 1.40}$ |
| FNO | $0.160_{\pm 0.004}$ | $1.08_{\pm 0.06}$ | $14.13_{\pm 0.34}$ | $0.128_{\pm 0.018}$ | $8.08_{\pm 0.15}$ | $5.61_{\pm 0.23}$ | $2.25_{\pm 1.14}$ | $7.40_{\pm 1.91}$ |
| WNO | $7.332_{\pm 0.307}$ | $2.23_{\pm 0.14}$ | $37.08_{\pm 1.23}$ | $0.572_{\pm 0.036}$ | $17.24_{\pm 0.46}$ | $12.05_{\pm 0.26}$ | $4.37_{\pm 0.08}$ | $22.22_{\pm 2.86}$ |
| SNO | $40.623_{\pm 8.437}$ | $8.55_{\pm 1.03}$ | $98.46_{\pm 0.25}$ | $94.891_{\pm 0.060}$ | $51.31_{\pm 0.01}$ | $62.34_{\pm 1.17}$ | $4.37_{\pm 0.87}$ | $21.93_{\pm 0.57}$ |
| DEEPONET | $10.561_{\pm 1.182}$ | $4.27_{\pm 0.24}$ | $55.48_{\pm 1.06}$ | $8.602_{\pm 0.431}$ | $24.59_{\pm 0.98}$ | $23.75_{\pm 0.20}$ | $2.85_{\pm 0.18}$ | $8.28_{\pm 0.37}$ |
| POD-DEEPONET | $3.999_{\pm 0.654}$ | $3.43_{\pm 0.04}$ | $33.37_{\pm 1.30}$ | $1.503_{\pm 0.145}$ | $29.63_{\pm 0.52}$ | $18.31_{\pm 1.17}$ | $4.14_{\pm 0.44}$ | $30.46_{\pm 0.59}$ |
| OFORMER | $0.165_{\pm 0.016}$ | $3.21_{\pm 0.06}$ | $10.97_{\pm 3.03}$ | $6.597_{\pm 0.352}$ | $27.33_{\pm 0.28}$ | $25.08_{\pm 1.36}$ | $41.75_{\pm 0.19}$ | $61.16_{\pm 0.49}$ |
| GNOT | $0.677_{\pm 0.021}$ | $2.04_{\pm 0.05}$ | $23.73_{\pm 0.97}$ | $0.102_{\pm 0.007}$ | $13.02_{\pm 0.81}$ | $9.99_{\pm 0.62}$ | $0.43_{\pm 0.02}$ | $0.71_{\pm 0.04}$ |
| LSM | $3.047_{\pm 0.434}$ | $1.10_{\pm 0.11}$ | $25.12_{\pm 0.12}$ | $0.377_{\pm 0.014}$ | $6.17_{\pm 0.23}$ | $4.07_{\pm 0.12}$ | $1.40_{\pm 0.12}$ | $7.11_{\pm 0.31}$ |

Table 1: Performance of different models across diverse datasets from distinct domains. The Relative L2 Error, expressed as ($\times 10^{-2}$), is the evaluation metric. Lower scores denote better performance. The optimal outcomes are highlighted in bold and dark blue, followed by the second-best in orange, and the third-best is underlined.

1. **BURGERS Equation**: This dataset models the one-dimensional flow of a viscous fluid. The input is the fluid's initial velocity distribution at time $t = 0$, and the output is the fluid's velocity at a time $t > 0$ Takamoto et al. (2022).

2. **DARCY Flow Equation**: The Darcy Flow dataset describes the steady-state flow of a fluid through a porous medium in two dimensions. The input is the spatial distribution of the medium's resistance to flow (viscosity), and the output is the fluid's velocity distribution across the domain at steady-state (Takamoto et al., 2022).

3. **NAVIER STOKES**: This dataset models the time evolution of a 2D viscous, incompressible fluid. The input includes the fluid's initial swirling motion (vorticity) and external forces acting on the fluid. The output is the fluid's velocity distribution over a specified time period (Takamoto et al., 2022).

4. **SHALLOW WATER Equation**: The shallow-water equations simulate the behavior of water that flows over a shallow surface in 2D. The input consists of the initial water depth and velocity distribution, and the output predicts the water flow dynamics in response to gravitational forces and varying underwater terrain (bathymetry) (Takamoto et al., 2022).

5. **STRESS Dataset**: This dataset models the stress distribution in a 2D binary composite material subjected to mode-I tensile loading. The input is the material microstructure (distribution of two materials), and the output is the stress field (STRESS) distribution of the digital composite (Mehran Rashid et al., 2022).

6. **STRAIN Dataset**: The strain dataset describes the deformation of a 2D binary composite material subjected to mode-I tensile loading. The input is the material microstructure, and the output is the resulting strain fields (STRAIN) (Mehran Rashid et al., 2022).

7. **SHEAR Dataset**: Part of the mechanical MNIST collection, this dataset simulates the deformation of a heterogeneous material block when forces are applied parallel to its surface (SHEAR). The input is the material microstructure, and the output captures element-wise displacements subjected to shear loading (Lejeune, 2020).

8. **BIAXIAL Dataset**: Another subset of the mechanical MNIST experiments, this dataset models the material's response when stretched equally in two perpendicular directions (equibiaxial loading). The input is the material microstructure, and the output records the full field displacement under BIAXIAL stretching (Lejeune, 2020).

9. **ELASTICITY Dataset**: For this dataset, an external tension is exerted on an incompressible material featuring an arbitrary void at its center. The input to the system is characterized by the material's structure, which is highly irregular and provided in the form of point clouds. The output is the inner stress within the material. (Li et al., 2022b).

10. **AIRFOIL Dataset**: The dataset characterizes the transonic flow of a fluid over an airfoil. Input to the system comprises the locations of mesh points configured in an irregularly structured mesh. The output corresponds to the captured Mach number associated with these specific locations.. (Li et al., 2022b).

## 5 BENCHMARKING RESULTS

We present the results of rigorous experimentation on PDE solvers across six tasks, each designed to elucidate the unique capabilities and strengths of the models. The diversity of the selected PDEs, sourced from (Takamoto et al., 2022), (Lejeune, 2020), (Li et al., 2022b) and (Mehran Rashid et al., 2022), encompasses both time-dependent and time-independent challenges, capturing the intrinsic computational complexity inherent to these tasks. Additionally, irregular grid datasets are included to evaluate the models' capabilities in handling datasets with real-life, general geometries, as opposed to uniform grids. The experiments conducted on novel mechanical datasets not previously encountered by the solvers offer invaluable insights for the broader scientific community.

In alignment with established experimental protocols, the dataset was split as follows: $\sim 80\%$ for training, $\sim 10\%$ for validation, and testing each; refer 7 for more details. We ensured a level playing field for each operator by defining a hyperparameter range and selecting the best subset for experimentation (see Table 8). Model optimization was achieved using Adam (Kingma & Ba, 2014) and AdamW (Loshchilov & Hutter, 2017) optimizers. Depending on the specific task, we employed either step-wise or cycle learning rate scheduling (Smith & Topin, 2019).

The training was conducted under optimal hyperparameter configurations, introducing variability through distinct random seeds and data splits. All experiments adhered to a fixed batch size of 20 and were executed on $1 \sim 8$ NVIDIA A6000 GPUs, with memory capacities of 48 GBs. To ensure fairness and accuracy in results, each experiment was replicated thrice with different seeds, and we report the mean and deviation in Relative L2 Error.

### 5.1 ACCURACY

Tab. 1 shows the performance of the models on the eight datasets. FNO architecture stands out on the majority of datasets. Subsequently, GNOT and LSM showcase exemplary performance on a significant proportion (5/8) of datasets. Similar results are observed in experiments conducted on irregular grid datasets; see Tab. 2.

| Datasets | Geo-FNO | DeepONet | POD-DeepONet | GNOT | OFormer | Geo-LSM |
|---|---|---|---|---|---|---|
| | | | Models | | | |
| ELASTICITY | $2.33_{\pm 0.16}$ | $10.14_{\pm 0.76}$ | $9.99_{\pm 0.08}$ | $\mathbf{1.27}_{\pm 0.04}$ | $1.85_{\pm 0.28}$ | $\underline{2.26}_{\pm 0.46}$ |
| AIRFOIL | $\underline{1.36}_{\pm 0.19}$ | $14.77_{\pm 0.30}$ | $12.07_{\pm 0.13}$ | $0.83_{\pm 0.06}$ | $2.33_{\pm 0.49}$ | $\mathbf{0.70}_{\pm 0.05}$ |

Table 2: Performance of different models on irregular grid datasets. The Relative L2 Error, expressed as $(\times 10^{-2})$, is presented. Only the models capable of handling irregular datasets are included.

FNO 's strength lies in its frequency space transformation. By capturing and transforming the lower frequencies present in the data, the FNO can approximate the solution operators of scientific PDEs. This approach, which uses the integral kernel in the Fourier space, facilitates a robust mapping between input and output function spaces, making it particularly adept at handling the complexities of the datasets in this study. GNOT employing a mixture-of-experts approach and its unique soft domain decomposition technique divides the problem into multiple scales, allowing it to capture diverse features of the underlying PDE. Each expert or head in the model focuses on a different aspect of the PDE, and their combined insights lead to a comprehensive understanding, especially for challenging datasets like SHEAR and BIAXIAL.

In contrast to other transformer-based approaches, LSM initially projects high-dimensional input data into a compact latent space, eliminating redundant information before learning the underlying partial differential equation (PDE). It utilizes a neural spectral block to learn the solution operator within this low-dimensional latent space, leveraging universal approximation capacity with favorable theoretical convergence guarantees. By employing attention for efficient data projection to and from the latent space and employing theoretically sound methods to learn the PDE from a lower-dimensional space, LSM consistently achieves low error rates. The OFORMER architecture that employs an innovative approach to solving spatio-temporal PDEs, exhibits best results in NAVIER STOKES dataset. It efficiently forwards the time step dynamics in the latent space by unrolling in the time dimension and initiating with a reduced rollout ratio. This method conserves significant space during training on time-dependent datasets while achieving high accuracy.

Among the models, only four inherently support datasets with irregular grids. To facilitate a comprehensive comparison, we introduce variants of LSM (GEO-LSM) and FNO (GEO-FNO). Notably, GNOT, tailored for practical applications involving irregular meshes, excels on both datasets. It utilizes MLP for initial data encoding into feature embeddings and leverages transformers to handle diverse input structures. While FNO and LSM demonstrate proficiency in handling uniform grid

| | Models | | | | | | | | | | | |
|---|---|---|---|---|---|---|---|---|---|---|---|---|
| Dataset Size | FNN | RESNET | UNET | cGAN | FNO | WNO | SNO | DEEPONET | POD-DEEPONET | OFORMER | GNOT | LSM |
| 25% | $4.80_{\pm 0.27}$ | $6.23_{\pm 0.23}$ | $\underline{2.60}_{\pm 0.14}$ | $3.28_{\pm 0.13}$ | $\mathbf{1.87}_{\pm 0.13}$ | $2.94_{\pm 0.20}$ | $24.70_{\pm 1.08}$ | $7.50_{\pm 0.45}$ | $5.09_{\pm 0.20}$ | $3.94_{\pm 0.13}$ | $3.61_{\pm 0.20}$ | $2.41_{\pm 0.22}$ |
| 50% | $3.95_{\pm 0.24}$ | $5.20_{\pm 0.29}$ | $\underline{2.10}_{\pm 0.11}$ | $2.54_{\pm 0.13}$ | $\mathbf{1.32}_{\pm 0.10}$ | $2.37_{\pm 0.18}$ | $24.70_{\pm 1.12}$ | $6.15_{\pm 0.41}$ | $4.17_{\pm 0.28}$ | $3.32_{\pm 0.08}$ | $2.70_{\pm 0.13}$ | $1.57_{\pm 0.16}$ |
| 100% | $3.47_{\pm 0.14}$ | $5.14_{\pm 0.23}$ | $2.10_{\pm 0.08}$ | $\underline{1.88}_{\pm 0.04}$ | $\mathbf{1.08}_{\pm 0.06}$ | $2.23_{\pm 0.14}$ | $8.55_{\pm 1.03}$ | $4.27_{\pm 0.24}$ | $3.43_{\pm 0.04}$ | $3.21_{\pm 0.06}$ | $2.04_{\pm 0.05}$ | $1.10_{\pm 0.11}$ |

Table 3: Data-Efficiency Analysis: The Relative L2 Error ($\times 10^{-2}$) is reported when trained with reduced subsets of 25% and 50% of the training dataset (left column). The testing and validation datasets remain consistent across all experiments.

| Dataset | | Models | | | | | | | | | | | |
|---|---|---|---|---|---|---|---|---|---|---|---|---|---|
| Train | Test | FNN | RESNET | UNET | cGAN | FNO | WNO | SNO | DEEPONET | POD-DEEPONET | OFORMER | GNOT | LSM |
| STRESS | STRESS | $25.69_{\pm 0.59}$ | $20.05_{\pm 0.19}$ | $10.57_{\pm 0.19}$ | $6.66_{\pm 0.84}$ | $\underline{8.08}_{\pm 0.15}$ | $17.24_{\pm 0.46}$ | $51.31_{\pm 0.01}$ | $24.59_{\pm 0.98}$ | $29.63_{\pm 0.52}$ | $27.33_{\pm 0.28}$ | $13.02_{\pm 0.81}$ | $\mathbf{6.17}_{\pm 0.23}$ |
| | STRAIN | $91.11_{\pm 0.04}$ | $\underline{89.83}_{\pm 0.79}$ | $95.79_{\pm 3.40}$ | $95.34_{\pm 1.57}$ | $95.39_{\pm 0.89}$ | $93.71_{\pm 4.97}$ | $\mathbf{62.36}_{\pm 0.46}$ | $92.70_{\pm 3.30}$ | $596.33_{\pm 23.70}$ | $68.70_{\pm 0.76}$ | $94.35_{\pm 1.09}$ | $96.31_{\pm 1.14}$ |
| STRAIN | STRAIN | $23.09_{\pm 7.08}$ | $14.64_{\pm 0.31}$ | $9.05_{\pm 0.33}$ | $\underline{6.12}_{\pm 0.80}$ | $5.61_{\pm 0.23}$ | $12.05_{\pm 0.26}$ | $62.34_{\pm 1.17}$ | $23.75_{\pm 0.20}$ | $18.31_{\pm 1.17}$ | $25.08_{\pm 1.36}$ | $9.99_{\pm 0.62}$ | $\mathbf{4.07}_{\pm 0.12}$ |
| | STRESS | $75.63_{\pm 1.49}$ | $\underline{77.29}_{\pm 0.72}$ | $77.41_{\pm 0.93}$ | $79.49_{\pm 1.07}$ | $79.50_{\pm 0.86}$ | $80.56_{\pm 1.27}$ | $\mathbf{51.65}_{\pm 1.10}$ | $77.49_{\pm 1.50}$ | $86.32_{\pm 2.24}$ | $80.26_{\pm 0.81}$ | $80.24_{\pm 0.95}$ | $80.35_{\pm 0.94}$ |

Table 4: Out-of-Distribution Evaluation: Models are trained on the STRESS dataset and subsequently tested on both the STRESS dataset and the out-of-distribution STRAIN dataset. The experiment is reciprocated with STRAIN as the training set. Relative L2 Error ($\times 10^{-2}$) is reported.

datasets, their effectiveness is maintained when an additional geometric layer learns the transformation from irregular input domains to a uniform transformed domain. However, this approach has limitations, particularly in tasks where efficient transformation from the input grid to a uniform grid space is challenging to learn.

Interestingly, most models, with the notable exception of GNOT, struggle to accurately learn the underlying PDE for the BIAXIAL and SHEAR datasets. The simpler FNN architecture demonstrates significant proficiency in learning these datasets. This underscores the versatility of such architectures, even when they aren't explicitly designed as operators.

## 5.2 DATA EFFICIENCY

We utilized the DARCY dataset with 1700 samples of $47 \times 47$ dimensions for the data-efficiency experiments. To assess the data efficiency of the models, we trained all models on reduced subsets: 25% (425 samples) and 50% (850 samples) of the original dataset.

The exceptional performance of frequency-based methods, notably FNO and WNO, even with limited data, is rooted in their operation within the frequency domain (see Table 3). The notable capability of these methods to capture the essential dynamics of the underlying PDEs through the lower frequencies present in the data enables data-efficient learning, a crucial feature for realistic data where the number of observations may be limited. Transformer-based neural operator architectures have demonstrated potential in approximating operators. However, their efficacy diminishes when data is sparse. GNOT, which typically excels with a rich dataset, struggles to outperform even basic neural network architectures in a data-limited scenario. On the other hand, LSM consistently remains the second best model in terms of error rates. However, more than a two-fold increase in the error shows the data dependence of the attention-based projection method used within the model. This trend underscores the inherent data dependency of transformer architectures, highlighting the challenges faced by many models, except frequency-based operators, when trained on limited data.

## 5.3 ROBUSTNESS TO NOISE

In practical applications, it's common to encounter noise in measurements. We simulated conditions with noisy data to understand how various neural operators handle such real-world challenges. We intentionally introduced corrupted input function data to each model during our testing phase. The goal was to see how well these models could still predict the ground truth amidst this noise.

Figure 3 shows the performance of the models on noisy data. Transformer-based architectures have shown commendable performance on the DARCY dataset. Even when noise is introduced, these models continue to perform well. However, the resilience of OFORMER and GNOT is tested when faced with the STRESS dataset. In scenarios where they already find it challenging to learn the underlying PDEs, the addition of noise exacerbates their performance issues.

On the other hand, SNO shows superior robustness to noise. While its performance in a noise-free environment is far from the best, it performs remarkably when exposed to noisy datasets, especially on the STRESS dataset. This resilience can be attributed to its unique approach: unlike other

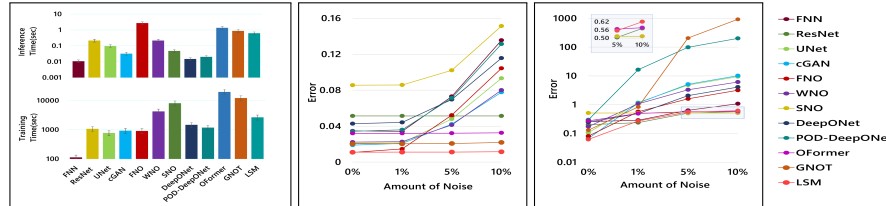

Figure 3: Time Efficiency: (On left) We report the time spent by each model during training (in the bottom) and inference time on the test set (on top). Results are collected during the training on the DARCY dataset. Robustness Analysis Against Noise: In terms of Relative L2 Error, performance metrics are presented for models subjected to random Gaussian noise. The evaluation encompasses the DARCY dataset (in the middle) and the STRESS dataset (on the right). The right diagram provides a detailed comparison snapshot of the most noise-resilient models on the STRESS dataset.

frequency-based methods that transition between the time and frequency domains, SNO exclusively processes data in the frequency domain. This design choice allows it to filter out noise, identifying it as a high-frequency disturbance before it begins its prediction process. A similar scenario is evident in the performance of LSM, as it remains largely unaffected by noisy input when tested on the Darcy dataset. Even when subjected to the demanding STRESS dataset, LSM consistently ranks among the best-performing models. This resilience is attributed to its projection into a compact latent space, eliminating redundant information, including the noise introduced from the coordinate space.

## 5.4 ZERO-SHOT SUPER-RESOLUTION

Directly approximating the solution operator offers a theoretical advantage: the potential for a mesh invariant continuous dynamical system. Once trained, such a system can ideally maintain accuracy even when applied to larger systems than those it was trained on. This capability is termed "zero-shot super-resolution." Note that FNO and GNOT enables zero shot super-resolution without any modifications. Upon closer examination, other models, such as SNO and DEEPONET, cannot have a straightforward application on zero-shot super-resolution. Instead, they lean on certain workarounds to achieve the desired results.

|  |  | Models | |
|---|---|---|---|
| Dataset | Resolution | FNO | GNOT |
| DARCY | **47 × 47** | **1.08**$_{\pm 0.06}$ | **2.04**$_{\pm 0.05}$ |
|  | 64×64 | 60.50$_{\pm 5.49}$ | 55.32$_{\pm 5.65}$ |
|  | 128×128 | 59.99$_{\pm 5.48}$ | 55.42$_{\pm 5.68}$ |
| STRAIN | **48 × 48** | **5.61**$_{\pm 0.23}$ | **9.99**$_{\pm 0.62}$ |
|  | 104×104 | 16.92$_{\pm 0.36}$ | 20.26$_{\pm 0.65}$ |
|  | 200×200 | 18.74$_{\pm 0.24}$ | 20.89$_{\pm 0.20}$ |

Table 5: Zero-shot super-resolution. Comparing various resolutions (left) with corresponding model performance (right). The original dataset resolution is highlighted.

While these modifications might enable super-resolution in practice, they diverge from the concept of zero-shot super-resolution from an architectural perspective. Accordingly, we consider only FNO and GNOT for our evaluation. Tests on both DARCY and STRAIN datasets are conducted, with the original training data having a lower resolution, and the neural operators are evaluated on higher resolution data. The inherent architecture of FNO fails to respect the continuous-discrete equivalence, leading to aliasing errors (Bartolucci et al., 2023). These errors exacerbate as the test data resolution differs from the training data; see Tab 5. Although GNOT performs slightly better, its results are still suboptimal. Further theoretical research is necessary to comprehend the mathematical foundations of learning a discretized version of PDE using transformer-based architectures and the limitation and scope of improvement regarding generalization capability to continuous domains.

## 5.5 OUT-OF-DISTRIBUTION GENERALIZATION

The equations for STRESS and STRAIN are intrinsically linked, differing primarily by the coefficient of elasticity, commonly known as Young's modulus. Given that our training and testing processes utilize normalized data, it's reasonable to anticipate that the models trained on the STRESS dataset should be adept at predicting strain in the material microstructures and vice versa. This expectation is particularly true for neural operators that grasp the underlying partial differential equations (PDEs) governing such relationships. Table 4 shows the OOD evaluation on all the models. Interestingly, for SNO, the error on the strain test dataset remains consistent, whether it was trained on the strain or stress datasets. The same holds when tested on the stress dataset. This consistency underscores SNO's ability to learn the underlying PDE. In stark contrast, other models don't exhibit this adaptability. Their accuracy levels decline when the testing set is swapped, indicating a potential limitation in their ability to generalize across closely related tasks.

## 5.6 Time Efficiency

Neural Operators are gaining traction over traditional numerical solvers due to their promise of rapid inference once trained. For this assessment, we've bench-marked various continuous dynamical systems on two criteria: the duration required to train on the DARCY dataset and the time needed to predict output function values for 200 test samples, each mapped on a uniform $47 \times 47$ grid. As anticipated, the FNN, with its straightforward architecture, stands out by requiring the least time for training and inference. However, when we delve into the other models, those based on deep operator regression methods show training duration on par with some complex but standard neural network architectures. For better visualization, see Figure 3. When considering inference time, the narrative shifts a pivotal metric in practical applications. While FNO is relatively efficient during training, it, along with the transformer-based models, takes a longer inference stride. While LSM outperforms other transformer-based architectures in both metrics, in the broader context of assessed operators, it doesn't rank as a highly time-efficient model in either training or inference time. Though all these models show promising performance on different metrics, inference time efficiency remains challenging. In stark contrast, most other models edge closer to offering real-time inference, highlighting the inherent time complexity trade-offs one must consider when opting for a particular neural operator.

## 6 Concluding Insights

The key insights drawn from this work are as follows.

1. **Operator and transformer**: FNO, GNOT and LSM emerge as the superior models across various metrics, suggesting that the architectural novelty in these models can indeed capture the maps between infinite-dimensional functional spaces.

2. **Spectral resilience to noise and OOD**: Despite its underwhelming performance, SNO exhibits superior adaptability to related datasets and resilience against noise. Similarly, SNO expresses impressive results on out-of-distribution datasets as well. The merit lies in its singular Fourier and inverse Fourier transformations, mapping input to output entirely in the frequency domain.

3. **Attention alone is not enough**: OFORMER, employing the attention-based transformer architecture, showcases notable advantages on the NAVIER STOKES dataset. It also demonstrates commendable results on specific PDEs like BURGERS, DARCY. However, a glaring limitation surfaces when these architectures are applied to other PDEs, whether of comparable complexity or even simpler ones. They fail to generalize. This shortcoming starkly contrasts with one of the primary advantages anticipated from data-driven PDE solvers: the capacity to discern the solution operator solely from data, independent of prior knowledge of the underlying PDE.

4. **Data-driven models work**: Surprisingly, the CGAN, a standard architecture for image tasks, excels in performance, even though it isn't inherently an operator. This prowess, however, wanes during cross-dataset evaluations, underscoring the importance of truly learning the underlying PDE rather than merely excelling on a given dataset.

5. **Challenges with SHEAR and BIAXIAL Datasets**: The collective struggle of most operators with the SHEAR and BIAXIAL datasets underscores the importance studying complex deformation patterns. Specifically, it suggests clear and well-defined operator failure modes where future works can be focused.

6. **Time efficiency should be improved**: While the models give reasonable performance, they grapple with time efficiency. Significantly, the best-performing models, such as transformer-based architectures, are time-intensive during training and inference, FNO is relatively swift in training but still intensive in inference.

**Limitations and Future Work:** Although FNO, GNOT and LSM exhibit consistent superior results, their results in cross-dataset evaluations and zero-shot super-resolution raise the questions of whether they are truly learning approximate solutions to the underlying PDE (see App. G). Similarly, although resilient to noise and OOD, the internal neural network architecture SNO remains largely unexplored and often yields subpar outcomes. Future endeavors leveraging SNO might pave the way to operators with improved robustness. Failure modes of operators in datasets require further investigations to build more robust operators that can capture complex shear deformations. Finally, the model's inference time requires improvement to be applied to large-scale real-world problems.

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

# A APPENDIX

## A.1 DATASET DETAILED MATHEMATICAL DESCRIPTION

### A.1.1 BURGERS EQUATION

The 1D Burgers' equation is a non-linear PDE which is having the following form:

$$\frac{\partial u}{\partial t}(x,t) + \frac{\partial}{\partial x}\left(\frac{u^2(x,t)}{2}\right) = \nu\frac{\partial^2 u}{\partial x^2}(x,t), \quad x \in (0,1),\ t \in (0,1] \tag{2}$$

with periodic boundary conditions, where $u(x,0) = u_0(x)$, $x \in (0,1)$.

Here, $u_0 \in L^2_{\text{per}}((0,1);\mathbb{R})$ is the initial condition and $\nu \in \mathbb{R}^+$ is the viscosity coefficient.

### A.1.2 DARCY FLOW EQUATION

We experiment with the steady-state solution of 2D Darcy Flow over the unit square, whose viscosity term $a(x)$ is an input of the system. The following equation defines the solution of the steady state.

$$\nabla \cdot (a(x)\nabla u(x)) = f(x), \quad x \in (0,1)^2, \tag{3}$$

$$u(x) = 0, \quad x \in \partial(0,1)^2. \tag{4}$$

For this paper, the force term $f$ is set to a constant value $\beta$, changing the scale of the solution $u(x)$. Instead of directly solving Equation 3, we get the solution by solving a temporal evolution equation:

$$\partial_t u(x,t) - \nabla \cdot (a(x)\nabla u(x,t)) = f(x), \quad x \in (0,1)^2, \tag{5}$$

The equation is solved with random initial conditions until it reaches a steady-state solution.

### A.1.3 COMPRESSIBLE NAVIER STOKES

We experimented with the 2-D Navier-Stokes equation for a viscous, incompressible fluid in vorticity form on the unit torus having the following form:

$$\partial_t w(x,t) + u(x,t) \cdot \nabla w(x,t) = \nu\Delta w(x,t) + f(x), \quad x \in (0,1)^2,\ t \in (0,T] \tag{6}$$

$$\nabla \cdot u(x,t) = 0, \quad x \in (0,1)^2,\ t \in [0,T] \tag{7}$$

$$w(x,0) = w_0(x), \quad x \in (0,1)^2 \tag{8}$$

where $u \in C([0,T]; H^r_{\text{per}}((0,1)^2;\mathbb{R}^2))$ for any $r > 0$ is the velocity field, $w = \nabla \times u$ is the vorticity, $w_0 \in L^2_{\text{per}}((0,1)^2;\mathbb{R})$ is the initial vorticity, $\nu \in \mathbb{R}^+$ is the viscosity coefficient, and $f \in L^2_{\text{per}}((0,1)^2;\mathbb{R})$ is the forcing function.

### A.1.4 INCOMPRESSIBLE NAVIER STOKES

The incompressible Navier-Stokes equations represent a specialized form of the broader compressible fluid dynamics equations tailored for subsonic flow scenarios. These equations are versatile and can be employed to analyze a wide range of systems, from hydromechanical processes to meteorological predictions and exploring turbulent behaviors.

### A.1.5 SHALLOW WATER EQUATION

The shallow-water equations present a suitable framework for modeling free-surface flow problems. Its 2D hyperbolic PDEs with the following form:

$$\partial_t h + \nabla \cdot (hu) = 0, \tag{9}$$

$$\partial_t hu + \nabla \cdot \left(u^2 h + \frac{1}{2}grh^2\right) = -grh\nabla b, \tag{10}$$

where $u$ and $v$ are the velocities in the horizontal and vertical directions, $h$ denotes the water depth, and $b$ describes spatially varying bathymetry. $hu$ represent as the directional momentum components and $g$ represent the gravitational acceleration.

| | Physical Quantity | |
|---|---|---|
| Dataset | Input | Output |
| BURGERS | Initial Velocity | Velocity |
| DARCY | Viscosity | Velocity |
| NAVIER STOKES | Vorticity | Velocity |
| SHALLOW WATER | Velocity | Water Flow Dynamics |
| STRESS | Material Microstructure | STRESS Field |
| STRAIN | Material Microstructure | STRAIN Field |
| SHEAR | Material Microstructure | Displacement under SHEAR loading |
| BIAXIAL | Material Microstructure | Displacement under BIAXIAL stretching |
| ELASTICITY | Mesh Point Location | Inner STRESS |
| AIRFOIL | Mash Point Location | Mach Number |

Table 6: Input and Output Physical Quantities modeled by each dataset.

# B MECHANICAL DATASETS

## B.1 STRESS, STRAINS FIELDS IN 2D DIGITAL IN COMPOSITES

The constitutive relationship (generalized Hook's law) is defined as :

$$\{\sigma_{ij}\} = \mathbb{C}_{ijkl}\{\epsilon_{kl}\} \tag{11}$$

where $\sigma_{ij}$ and $\epsilon_{kl}$ are the stress and strain components, $\mathbb{C}_{ijkl}$ is the overall stiffness tensor. For 2D problems, we have $\sigma_{xx}$ , $\sigma_{yy}$ and $\sigma_{xy}$ satisfy the following equilibrium equations

$$\partial_x\sigma_{xx} + \partial_y\sigma_{xy} + F_x = 0; \partial_y\sigma_{yy} + \partial_x\sigma_{xy} + F_y = 0 \tag{12}$$

where $F_x$ and $F_y$ are the body forces in horizontal and vertical directions, respectively. Additionally, the strains $\epsilon_{xx}$, $\epsilon_{yy}$ and $\epsilon_{xy}$ are defined as

$$\epsilon_{xx} = \partial_x u_{xx}; \epsilon_{yy} = \partial_y u_{yy}; \epsilon_{xy} = \partial_y u_x + \partial_x u_x \tag{13}$$

where $u_x$ and $u_y$ are the displacements in horizontal and vertical directions, respectively
The 2D-Digital composites(Mehran Rashid et al., 2022) are subjected to mode-I tensile loading with specific boundary conditions. The two-phase composite has a modulus ratio of 10 $(E_{stiff}/E_{soft})$, and both materials are assumed to be perfectly elastic. The simulations are run on an $8mm \times 8mm$ plate with negligible thickness. The two phases are randomly placed over the plate; however, the fraction of soft and stiff phases is equal.

## B.2 MECHANICAL MNIST

The mechanical MNIST (Lejeune, 2020) has a collection of different tests such as shear, equibiaxial extension, confined compression, and other tests simulated on a heterogenous material subjected to large deformations. The famous MNIST digit dataset is generated by treating the bitmap images as heterogeneous material blocks modeled as a Neo-Hokean material of varying modulus. The material is subjected to fixed displacements in one direction, and the full-field displacements are recorded at each step. In this study, we select two tests from the mechanical MNIST experiments: shear and equibiaxial.

# C RESULT GENERATION SETTING

We include detailed description (see Table 7) about the datasets and their specific configurations (resolution, data split, time steps in input/output). This can be used for reproducibility of the results.

| Dataset | Grid Resolution | Time Steps | | Dataset Split | | | |
| --- | --- | --- | --- | --- | --- | --- | --- |
| | | Input | Output | Total Samples | Training Data Size | Validation Data Size | Testing Data Size |
| BURGERS | (8192) | | | 2048 | 1700 | 148 | 200 |
| DARCY | $(47 \times 47)$ | | | 2000 | 1700 | 100 | 200 |
| NAVIER STOKES | $(47 \times 47)$ | 5 | 16 | 1200 | 900 | 100 | 200 |
| SHALLOW WATER | $(47 \times 47)$ | 10 | 40 | 1000 | 800 | 100 | 100 |
| STRESS | $(48 \times 48)$ | | | 1200 | 900 | 100 | 200 |
| STRAIN | $(48 \times 48)$ | | | 1200 | 900 | 100 | 200 |
| SHEAR | $(28 \times 28)$ | | | 70000 | 50000 | 10000 | 10000 |
| BIAXIAL | $(28 \times 28)$ | | | 70000 | 50000 | 10000 | 10000 |

Table 7: This table contains descriptions of datasets used with information about the data resolution, number of time steps, and dataset split.

## D  HYPERPARAMETERS FOR TRAINING MODELS

We include the choice of optimizer, scheduler, learning rate that gives best results for each of the model on DARCY flow dataset (refer to Table 8). Number of parameters used by each model are also included to give further details in terms of model complexity.

| Model | Optimizer | Scheduler | Learning Rate | Number of Parameters |
| --- | --- | --- | --- | --- |
| FNN | Adam | StepLR | 0.0001 | 2330273 |
| RESNET | Adam | StepLR | 0.001 | 28060769 |
| UNET | Adam | StepLR | 0.0001 | 31031745 |
| cGAN | Adam | OneCycleLR | 0.0001 | 2240963 |
| FNO | Adam | StepLR | 0.001 | 1188353 |
| WNO | Adam | StepLR | 0.001 | 5337985 |
| SNO | AdamW | StepLR | 0.001 | 147321 |
| DEEPONET | AdamW | StepLR | 0.0001 | 36465408 |
| POD-DEEPONET | AdamW | StepLR | 0.0001 | 36413811 |
| OFORMER | Adam | OneCycleLR | 0.001 | 1476481 |
| GNOT | AdamW | OneCycleLR | 0.0001 | 6451201 |
| LSM | Adam | StepLR | 0.0001 | 4806657 |

Table 8: Best Hyperparameters Used for Training the Models On DARCY Dataset, Hyperparameters For Other Datasets Can Be Found Within The Codebase
.

## E  RELATED WORK

In our comparative analysis, CODBENCH stands out among existing repositories for its comprehensive inclusion of diverse architectures in the domain of scientific PDEs and data-driven PDE solvers. The detailed results of this comparison are presented in Table 9. We specifically consider four repositories that offer a rich variety of resources, ensuring a fair assessment of CODBENCH's functionality.

- **DeepXDE:** **(Lu et al., 2021b)**
- **PDEBench:** **(Takamoto et al., 2022)**
- **BubbleML:** **(Hassan et al., 2023)**
- **CFDBench:** **(Luo et al., 2023)**

## F  TRAINING DETAILS

All models undergo tuning of the learning rate through a grid search method, selecting rates from the set $\{10^{-4}, 10^{-3}, 10^{-2}\}$. In cases of unstable training trajectories across all rates in the initial set, ad-

| Model | Codebase | | | | |
|---|---|---|---|---|---|
| | DeepXDE | PDEBench | BubbleML | CFDBench | CoDBench |
| FNN | ✓ | ✗ | ✗ | ✓ | ✓ |
| CNN | ✓ | ✗ | ✗ | ✓ | ✓ |
| UNet | ✗ | ✓ | ✓ | ✓ | ✓ |
| cGAN | ✗ | ✗ | ✗ | ✗ | ✓ |
| FNO | ✗ | ✓ | ✓ | ✓ | ✓ |
| WNO | ✗ | ✗ | ✗ | ✗ | ✓ |
| DeepONet | ✓ | ✗ | ✗ | ✓ | ✓ |
| POD-DeepONet | ✓ | ✗ | ✗ | ✗ | ✓ |
| SNO | ✗ | ✗ | ✗ | ✗ | ✓ |
| OFormer | ✗ | ✗ | ✗ | ✗ | ✓ |
| GNOT | ✗ | ✗ | ✗ | ✗ | ✓ |
| LSM | ✗ | ✗ | ✗ | ✗ | ✓ |

Table 9: Detailed comparison of implementations provided by different neural operator codebases.

ditional experiments are conducted with rates from $\{10^{-5}, 10^{-1}\}$. Each experiment is executed once for every model-dataset pair. The learning rate yielding the best performance for a specific model on a given dataset is selected. Three experiments with different random seeds are then conducted using the optimal learning rate, and the mean and deviation of the results are reported. Table 10 displays the Relative L2 error achieved by the models when trained to convergence with three different learning rates. Note that most values lack a standard deviation as the initial learning rates are not optimal, and only one experiment is performed for each. However, the optimal learning rate includes both mean and standard deviation, derived from three experiments using distinct seeds. All experiments are executed until convergence, which is determined by two predefined criteria. The patience is set to 100 epochs, and $\epsilon$ is set to 1e-6. Convergence is considered achieved when the validation set error fails to decrease by at least the quantity $\epsilon$ from the 'best validation error yet' for the patience number of epochs. Once this criterion is met, the training process is halted, and the model is tested on the test dataset.

| Dataset | LR | FNN | ResNet | UNet | cGAN | FNO | WNO | SNO | DeepONet | POD-DeepONet | OFormer | GNOT | LSM |
|---|---|---|---|---|---|---|---|---|---|---|---|---|---|
| | | | | | | | | Models | | | | | |
| Darcy | 0.01 | 9.04 | 5.73 | 37.18 | 30.75 | 1.12 | 5.39 | 24.65 | 24.61 | 8.44 | 5.01 | 6.93 | 6.25 |
| | 0.001 | 3.87 | $5.14_{\pm 0.23}$ | 5.37 | 3.02 | $1.08_{\pm 0.06}$ | $2.23_{\pm 0.14}$ | $8.55_{\pm 1.03}$ | 4.89 | 3.54 | $3.21_{\pm 0.06}$ | 2.57 | 3.56 |
| | 0.0001 | $3.47_{\pm 0.14}$ | 5.33 | $2.10_{\pm 0.08}$ | $1.88_{\pm 0.04}$ | 1.60 | 2.87 | 25.47 | $4.27_{\pm 0.24}$ | $3.43_{\pm 0.04}$ | 7.82 | $2.04_{\pm 0.05}$ | $1.10_{\pm 0.11}$ |

Table 10: Hyper-parameteric sensitivity: Results of models on 3 different learning rates when tested on DARCY Flow dataset. Relative L2 Error ($\times 10^{-2}$) is reported

## G  LIMITATIONS

The partial differential equations (PDEs), especially those governing fluid dynamics, can exhibit diverse behavior depending on the boundary conditions. Although we introduce an out-of-distribution generalization task with strain-stress datasets, further experiments varying the boundary conditions could provide valuable insights into each model's behavior and generalization capabilities. Despite including 12 different operators for the benchmarking study, achieving exhaustiveness is challenging. The list below highlights some interesting neural network architectures that were not incorporated in this study: Graph Kernel Neural Operator, Low Rank Neural Operator, Multipole Graph Neural Operator, Markov Neural Operator, and Laplace Neural Operator.

