# OpenReview forum: "CoDBench: A Critical Evaluation of Data-driven Models for Continuous Dynamical Systems"
_ICLR.cc/2024/Conference — ICLR 2024 Conference Withdrawn Submission_

### Official Review · Reviewer_M9NG · 2023-10-28

**Soundness:** 3 good
**Presentation:** 3 good
**Contribution:** 2 fair
**Rating:** 3
**Confidence:** 3

**Summary:**

This paper proposes an benchmarking suite comprising 11 state-of-the-art data-driven models for solving differential equations. Authors conduct extensive experiments, assessing the operators’ capabilities in learning, zero-shot super-resolution, data efficiency, robustness to noise, and computational efficiency.

**Strengths:**

1. Building benchmark is important for research community.
2. Authors conduct a lot of experiments.

**Weaknesses:**

1. I feel like more contributions should be made to meet the ICLR requirement. Authors only compare 11 methods in standard benchmarks. I suggest authors should add more methods or new datasets for a more comprehensive benchmark.
2. More observations and conclusion should be made throughout the experiments. Now the observations is lack of insights.
3. A lot of dynamical system modeling methods are missed for comparison, e.g., [1]. More video prediction methods can be also compared.

[1] Solving High-Dimensional PDEs with Latent Spectral Models ICML 23

**Questions:**

See above.

---

> ### Author Response · Authors · 2023-11-19
> **Response to comments by Reviewer M9NG**
>
> *Q. I feel like more contributions should be made to meet the ICLR requirement. Authors only compare 11 methods in standard benchmarks. I suggest authors should add more methods or new datasets for a more comprehensive benchmark.*
>
> **Response:** We thank the reviewer for their suggestion. We have now added an additional methods (LSM) and two additional datasets on irregular grids (Elasticity, Airfoil) in the experiments. The benchmark now consists of 12 benchmark models and 10 datasets. In addition, a new dataset is considered for the super-resolution evaluation.
>
> *Q. More observations and conclusion should be made throughout the experiments. Now the observations is lack of insights.*
>
> **Response:** Additional observations based on added super-resolution test and irregular grid datasets are added.
>
> *Q. A lot of dynamical system modeling methods are missed for comparison, e.g., [1]. More video prediction methods can be also compared.
> [1] Solving High-Dimensional PDEs with Latent Spectral Models ICML 23*
>
> **Response:** We have now included 12 models and 10 datasets for the benchmark experiments. LSM is added as an additional method (see results section for more details). Additionally, Geo-FNO and Geo-LSM are added as an upgrade for FNO and LSM models for performing experiments on the added irregular grid datasets.
>
> **Appeal to the reviewer:** With the additional experiments, results, and explanations, we now hope the reviewer finds the manuscript suitable for publication. Accordingly, we request you to raise the score for the manuscript. Please do let us know if there are any further queries.

---

> > ### Author Response · Authors · 2023-11-22
> > **Keenly awaiting post-rebuttal feedback**
> >
> > Dear Reviewer M9NG,
> >
> > As the author-reviewer discussion phase is approaching its conclusion in just a few hours, we are reaching out to inquire if any remaining concerns or points require clarification. Specifically, the following changes have been made.
> >
> > 1. Additional experiments on **irregular grids** on two new datasets and two new models (**geo-LSM and geo-FNO**).
> > 2. A new model, namely, **LSM**, suggested by the reviewer, is added for all the datasets.
> > 3. Additional experiment on the **strain dataset for superresolution**.
> > 4. Added a table on comparison of **CoDBench** to previous works (see response to Reviewer fbDR). It can be noted that CoDBench evaluated 12 different models, whereas the next best model in the literature has performed evaluations on only 5 different models.
> > 5. Finally, CoDBench is submitted as a **benchmarking paper** under the topic ``datasets and benchmarks'' in the ICLR paper track (https://iclr.cc/Conferences/2024/CallForPapers). We request the reviewer to evaluate it as a benchmarking paper rather than a paper proposing an original architecture.
> >
> > With these comments and modifications, we hope we have addressed all the outstanding concerns of the reviewer. We request you to raise any additional concerns you have. Otherwise, we would really appreciate it if you could raise your score for the paper.
> >
> > Your support in this final phase, particularly if you find the revisions satisfactory, would be immensely appreciated.

---

> ### Comment · Reviewer_M9NG · 2023-11-23
>
> Thanks for your response. I still think the contribution is not enough for ICLR benchmark. For example, how about more difficult scenarios like OOD scenarios? Moreover, more video prediction methods can also be compared with. I agree with "the scope of this contribution is limited". Also the observations is still lack of insights. I tend to keep my score.

---

### Official Review · Reviewer_fbDR · 2023-10-28

**Soundness:** 2 fair
**Presentation:** 2 fair
**Contribution:** 1 poor
**Rating:** 3
**Confidence:** 3

**Summary:**

The paper presents CoDBench, a computational benchmarking suite which contains 11 state-of-the-art data-driven models for solving differential equations. In addition, it provides eight benchmarking datasets for model evaluation. Using the computational suite the authors evaluate the models according to a division into four categories, according to their architecture. The comprehensive evaluation is used to draw key insights regarding the studied frameworks and datasets.

**Strengths:**

$\underline{\textrm{Originality}}$: instead of suggesting a novel approach this paper aims to analyze the strengths and potential applications of existing data-driven tools to study continuous dynamical systems. In a sense optimizing the usage of existing frameworks.

$\underline{\textrm{Quality}}$: the paper is well written. It provides a detailed introduction to the field of data driven approaches on PDEs as well as an elaborate description of benchmark results and conclusions.

$\underline{\textrm{Clarity}}$: all sections, from introduction to the concluding insights, are clearly presented.

$\underline{\textrm{Significance}}$: the significance of the paper is in providing a resource for exploration of dynamical systems as well as insights to optimal usage of assessed frameworks.

**Weaknesses:**

1. CoDBench package: the code package is presented as a major contribution which shall serve as a resource for studying dynamical systems. However, the code that is currently available (https://anonymous.4open.science/r/cod-bench-7525) is lacking a basic README with minimal guidelines for a user interested in using the package. It will be beneficial to accompany the code with proper documentation as well as detailed examples as common for many scientific code packages (see next point).
2. Related work: it would be beneficial for the authors to relate to existing code alternatives and or attempts to study data-driven models. For example, DeepXDE (lu et al. 2021) is a code package that contains most of the components of CoDBench and further allows more flexibility in terms of data construction.
3. Overall contribution: the authors present as major contributions the analysis of Super-resolution as well as out-of-distribution tasks (stress and strain) however both are presented in earlier works for studying data-driven models and benchmarked against baselines (e.g. Fanaskov et al. 2022 and Rashid et al. 2022). Hence while it is beneficial to analyze these at a broader scale over all 11 models the scope of this contribution is limited. Following this, the key insight, limitations, and future work suggested appear minimal in terms of scope and practicality.
4. Minor: Figure 1 is never referenced. It will be beneficial to change the coloring of optimal values in tables and or underline the second best to have better discrimination between second and third.

**Questions:**

1. Would it be possible to improve the provided package in light of the comment above?
2. Could the authors clarify what they see as the major contributions in light of the comments?

---

> ### Author Response · Authors · 2023-11-19
> **Response to comments by Reviewer fbDR (Part 1)**
>
> *CoDBench package: the code package is presented as a major contribution which shall serve as a resource for studying dynamical systems. However, the code that is currently available (https://anonymous.4open.science/r/cod-bench-7525) is lacking a basic README with minimal guidelines for a user interested in using the package. It will be beneficial to accompany the code with proper documentation as well as detailed examples as common for many scientific code packages (see next point).*
>
> **Response:** We thank the reviewer for raising this valuable point. We have now included detailed description about how to use the package. Working examples in form of run scripts are also included. Further reference to the run_scripts is included in the ReadME File. These additional information will allow readers and users to directly use the models and datasets to train and evaluate the architectures for different tasks.
>
> *Related work: it would be beneficial for the authors to relate to existing code alternatives and or attempts to study data-driven models. For example, DeepXDE (lu et al. 2021) is a code package that contains most of the components of CoDBench and further allows more flexibility in terms of data construction.*
>
> Response: We thank the reviewer for this suggestion, To the best of our knowledge, this is the only work that extensively evaluates most of the existing architectures for dynamical systems. While, in this work, we evaluate 12 architectures, the previous work [2] in the literature has evaluated a maximum of five or six architectures. Please see a comparison of the CODBench with other existing literature (including DeepXDE).
>
> [2] CFDBench: A Comprehensive Benchmark for Machine Learning Methods in Fluid Dynamics. (Arxiv: 13th September 2023)
>
> |              |         |          | CodeBases |          |                 |
> |--------------|---------|----------|-----------|----------|-----------------|
> | Models       | DeepXDE | PDEBench | BubbleML  | CFDBench | CoDBench (Ours) |
> | FNN          | &check;   | &cross;        | &cross;         | &check;       | &check;              |
> | ResNet       | &check;      | &cross;        | &cross;         | &check;       | &check;              |
> | UNet         | &cross;       | &check;       | &check;        | &check;       | &check;              |
> | cGAN         | &cross;       | &cross;        | &cross;         | &cross;        | &check;              |
> | FNO          | &cross;       | &check;       | &check;        | &check;       | &check;              |
> | WNO          | &cross;       | &cross;        | &cross;         | &cross;        | &check;              |
> | SNO           | &cross;       | &cross;        | &cross;         | &cross;        | &check;              |
> | DeepONet     | &check;      | &cross;        | &cross;         | &check;       | &check;              |
> | POD-DeepONet | &check;      | &cross;        | &cross;         | &cross;        | &check;              |
> | OFormer      | &cross;       | &cross;        | &cross;         | &cross;        | &check;              |
> | GNOT         | &cross;       | &cross;        | &cross;         | &cross;        | &check;              |
> | LSM          | &cross;       | &cross;        | &cross;         | &cross;        | &check;              |
>
> To put the present work in context, we have now added a table (see Table 9) in the Appendix which compares the present work with the literature in terms of architectures.

---

> > ### Author Response · Authors · 2023-11-19
> > **Response to comments by Reviewer fbDR (Part 2)**
> >
> > *The authors present as major contributions the analysis of Super-resolution as well as out-of-distribution tasks (stress and strain) however, both are presented in earlier works for studying data-driven models and benchmarked against baselines (e.g., Fanaskov et al. and Rashid et al.). Hence while it is beneficial to analyze these at a broader scale over all 11 models, the scope of this contribution is limited. Following this, the key insight, limitations, and future work suggested appear minimal in terms of scope and practicality.*
> >
> > **Response:** Super-resolution test is included in the SNO paper however it relies on workarounds for both sno and deeponet, as both, in a general theoretical way, can't do super-resolution. Also, the dataset used in sno papers is more mathematical than application-based missing the real-life application of super-resolution. Rashid et al. is very limited as:
> > 1. None of the papers include the shear and biaxial datasets for neural operators,
> > 2. Even the use of stress, strain datasets in the paper by Rashid et al. is limited to only the use of a few operators, whereas this paper tests 12 different operators and also includes an out-of-distribution task-based stress, strain dataset, which provides key insights into the learning behavior of each of the operators. Also, additional experiments on the strain dataset super-resolution have been added.
> >
> > *Figure 1 is never referenced. It will be beneficial to change the coloring of optimal values in tables and or underline the second best to have better discrimination between second and third.*
> >
> > **Response:** We thank the reviewer for this suggestion; we have added a reference to Figure 1; see the Introduction section for the change. We have also changed the coloring for better discrimination in all the tables.
> >
> > *Would it be possible to improve the provided package in light of the comment above?*
> >
> > **Response:** As outlined earlier, the package now contains all essential details to allow users to run experiments on all 12 provided operators. Multiple example scripts are included, which allow users to run all Table 1 experiments included in the paper.
> >
> > *Could the authors clarify what they see as the major contributions in light of the comments?*
> >
> > **Response:** The paper is intended to be a benchmark paper and hence conducts extensive experimentation in order to guide the readers in choosing the best method available for their particular task. The experiments conducted are well thought of as to compare the models on different aspects and applications of data-driven PDE solvers. One of the major contributions is the insights drawn from the experiments. The package provided is to be seen as an additional contribution as we provide very modular code that allows the scientific community to use 12 different operators only based on the latest machine learning framework PyTorch. See, the loader and trainer modules to further see the integration of all models to a common, easy-to-use framework for loading training and testing on datasets. The major contributions of the paper are as follows.
> > 1. The ability of each of the operators to learn on 10 different datasets provides insights into which architectures work better for learning PDEs and why.
> > 2. Additional experiments simulating real-life conditions to assess the usefulness and shortcomings of each of the operators. We conduct experiments for data efficiency, assessing the behavior when available data is scarce. We conduct experiments on noise efficiency, assessing whether operators can perform well on real-life datasets that might not be as perfect of data as created by simulation. We provide key insights into which models perform reasonably well in such real-life conditions. Additionally, the time efficiency of each model is a key factor in real-life applications. The study of time efficiency is also part of the real-life application capabilities of each of the models.
> > 3. We further evaluate the theoretical concept of an operator with these data-driven solvers to asses whether these operators are actually learning the underlying pde or just learning to fit the data. Theoretically, an operator can handle out-of-distribution tasks as the underlying PDE remains the same, as well as has the capability of doing zero-shot super-resolution. To the best of the authors' knowledge, the out-of-distribution tasks by switching the strain and stress datasets is a novel task introduced in this paper. Rashid et al. and any other work for that matter, doesn't include such experimentation.
> > 4. Last, but not least, we provide cod-bench, a seamless package to use 12 PDE solvers using the latest machine learning framework pytorch.
> >
> > **Appeal to the reviewer:** With the additional experiments, results, and explanations, we now hope the reviewer finds the manuscript suitable for publication. Accordingly, we request you to raise the score for the manuscript. Please do let us know if there are any further queries.

---

> > ### Comment · Reviewer_fbDR · 2023-11-21
> >
> > I would like to thank the authors for their response.
> > Unfortunately, it does not address my main concern, the contribution of this work, hence I will keep my score as is. In addition, while the number of evaluated architectures in this CoDBENCH may be greater the functionality it provides for model evaluation can not be compared to DeepXDE's (for example the latter includes data generation, domain formulation and supports multiple backends) hence I find the new table misleading in terms of comparison between the frameworks.

---

> ### Author Response · Authors · 2023-11-22
> **Thank you and further feedback**
>
> Dear Reviewer fbDR,
>
> Thank you for raising concern. We would like to highlight that CoDBench is directly comparable with DeepXDE as the former is a benchmarking paper, while the latter focuses on data generation, domain formulation, etc., as the reviewer outlines. The present work, CoDBench, is submitted as a benchmarking paper under the topic ``datasets and benchmarks'' in the ICLR paper track (https://iclr.cc/Conferences/2024/CallForPapers). Accordingly, we respectfully disagree with the comparison that the reviewer has made.
>
> The aim of the table, which was included based on the Reviewer's recommendation, was to show the exhaustive nature of CoDBench in benchmarking models; the present work benchmarks 12 models, while the next best work in the literature benchmarks only 5 models. We thank the reviewer for acknowledging this aspect.
>
> It should also be noted that based on the reviewer's comments, we have now added additional experiments on **irregular grids**, new models (LSM, geo-LSM, and geo-FNO), and superresolution. With these, we hope the reviewer finds the manuscript suitable for ICLR as a benchmarking paper.
>
> Since we are into the last few hours of author-reviewer discussion, we request the reviewer to raise any outstanding concerns. Otherwise, we earnestly request the reviewer to evaluate the paper for what it is and raise the score appropriately. Your support in this final phase, particularly if you find the revisions satisfactory, would be immensely appreciated!
>
> Thank you!

---

> > ### Comment · Reviewer_fbDR · 2023-11-23
> >
> > In the evaluation of the paper of _what it is_, and taking into consideration the reviews given by fellow reviewers I choose to keep my score as is.

---

### Official Review · Reviewer_3R3p · 2023-11-01

**Soundness:** 4 excellent
**Presentation:** 4 excellent
**Contribution:** 2 fair
**Rating:** 6
**Confidence:** 3

**Summary:**

This work proposes an exhaustive study of several machine learning algorithms of various kind (neural operators, frequency based decomposition, and more traditional approaches) on several types of dataset involving both fluid dynamics and mechanics.

The study not only focuses on the prediction capabilities of the tested algorithms but also studies their ability in selected related tasks such as super resolution and out-of-distribution prediction.

This work cannot be considered with high technical contributions yet it depicts an interesting snapshot of the current literature on several tasks related to dynamical system estimation and is in that sense valuable.

**Strengths:**

This work is a systematic study providing practitioners with a clear view of the tested algorithms capacities to fit different problems and what limit one should expect when tackling another task.

The presentation is clear and the paper is easy to follow and the auxiliary tasks are in my opinion interesting and valuable to test the model capacities.

**Weaknesses:**

The main limitation of the present paper is the lack of technical novelty and limited novel contribution to the field.

Yet, such a paper can define an interesting milestone of the field depending on the release and quality of the code and dataset provided alongside the paper.

For other concerns see questions.

**Questions:**

1. What training methodology is used for each algorithm ? I find section 2.3 quite general and it is difficult to understand what physical quantity is estimated / trained on by each algorithm.

2. Can the authors comment on the limitation of their study for the ood evaluation. PDEs solution can exhibit very different behavior with the same equation varying the initial condition, notably for fluid dynamics data. Such a test could strengthen the analysis proposed by the authors.

3. The error in the super resolution settings seems to explode. Can the authors comment on the results they report in the paper ?

4. While it is very difficult to be exhaustive in such a study, testing all the literature is unfeasible, I strongly encourage the authors to include an extensive discussion on why they chose the selected models or at least why they chose not to select other models. Such a discussion would be a valuable addition to the paper.

---

> ### Author Response · Authors · 2023-11-19
> **Response to the comments of Reviewer 3R3p**
>
> *The main limitation of the present paper is the lack of technical novelty and limited novel contribution to the field. Yet, such a paper can define an interesting milestone of the field depending on the release and quality of the code and dataset provided alongside the paper.*
>
> **Response:** We would like to bring to the attention of the reviewer is that this work is aimed at benchmarking the architectures for simulating dynamical systems. To the best of our knowledge, this is the only work that extensively evaluates most of the existing architectures for dynamical systems. While prior research, limited to [2], has not explored architectures beyond a certain scope, our evaluation includes 12 distinct architectures. Notably, existing literature typically assesses no more than five or six architectures. To provide a comprehensive perspective in comparison to both benchmarking studies and repositories featuring implementations for various neural operators, we have introduced Table 9 in the Appendix. This table provides a comparative analysis between our work and existing literature, specifically focusing on architectural considerations.
>
> [2] CFDBench: A Comprehensive Benchmark for Machine Learning Methods in Fluid Dynamics. (Arxiv: 13th September 2023)
>
> *What training methodology is used for each algorithm ? I find section 2.3 quite general and it is difficult to understand what physical quantity is estimated / trained on by each algorithm.*
>
> **Response:** To clarify this point, we have now included an additional table (Table 6) in the appendix that specifies the input and output physical quantity associated with each of the dataset.
>
> *Can the authors comment on the limitation of their study for the ood evaluation. PDEs solution can exhibit very different behavior with the same equation varying the initial condition, notably for fluid dynamics data. Such a test could strengthen the analysis proposed by the authors.*
>
> **Response:** We thank the reviewer for this suggestion. The PDEs can indeed exhibit very different behavior based on the boundary conditions set, especially for fluid dynamics data. Even though, we have added an out-of-distribution generalization task in terms of strain-stress datasets, Adding more experiments by varying the boundary conditions and studying the behavior of each model and the ability to generalize can indeed give further helpful insights. We have added this in the limitations section (Appendix G).
>
> *The error in the super resolution settings seems to explode. Can the authors comment on the results they report in the paper ?*
>
> **Response:** We have now added additional experiment testing the super-resolution capability of models on the strain dataset (see Tab. 5). Based on the results on both the datasets we have added additional insights on why such behavior is seen during super-resolution.
>
> *While it is very difficult to be exhaustive in such a study, testing all the literature is unfeasible, I strongly encourage the authors to include an extensive discussion on why they chose the selected models or at least why they chose not to select other models. Such a discussion would be a valuable addition to the paper.*
>
> **Response:** We thank the reviewer for this suggestion, and have added an extensive discussion on how models are selected. We have also added discussion about some of the models that have given good results but couldn't be included in the study as part of limitations.
>
> **Appeal to the reviewer:** With the additional experiments, results, and explanations, we now hope the reviewer finds the manuscript suitable for publication. Accordingly, we request you to raise the score for the manuscript. Please do let us know if there are any further queries.

---

> > ### Author Response · Authors · 2023-11-22
> > **Awaiting post-rebuttal feedback**
> >
> > Dear Reviewer 3R3p,
> >
> > As the author-reviewer discussion phase is approaching its conclusion in just a few hours, we are reaching out to inquire if any remaining concerns or points require clarification. Specifically, the following changes have been made.
> >
> > 1. Additional experiments on **irregular grids** on two new datasets and two new models (geo-LSM and geo-FNO).
> >
> > 2. A new model, namely, **LSM**, is added for all the datasets.
> >
> > 3. Additional experiment on the **strain dataset for superresolution**.
> >
> > 4. Further clarifications on the hyperparametric optimization.
> >
> > With these, we hope we have addressed all the outstanding concerns of the reviewer. We request you to raise any additional concerns you have. Otherwise, we would really appreciate it if you could raise your score for the paper.
> >
> > Your support in this final phase, particularly if you find the revisions satisfactory, would be immensely appreciated.

---

### Official Review · Reviewer_eLXu · 2023-11-06

**Soundness:** 3 good
**Presentation:** 3 good
**Contribution:** 3 good
**Rating:** 6
**Confidence:** 2

**Summary:**

This paper provides a benchmark for data-driven dynamic modeling. Evaluating 4 types of the model - feed-forward neural networks, deep operator regression models, frequency-based neural operators, and transformer architectures, on 8 benchmark datasets - evaluate the robustness to noise, computational efficiency, and data efficiency of the model, with opensource data and codebase.

**Strengths:**

This work performs quite a comprehensive quantitative analysis of different models on a wide range of datasets, testing model capability over data efficiency, run time, training time, prediction accuracy, and super-resolution accuracy. It is nice to have a benchmark of different model on regular girds.

**Weaknesses:**

This is a benchmark paper on simulation on a regular grid. I am not sure if I missed this, the discussion on performance/adaptability of different models on irregular grids is not included. I found the overall analysis and insights comprehensive but also a little bit simple with missing key discussion on a simulation where irregular grids are required.

**Questions:**

I am curious about more details on how hyperparameters are set for different model training across dataset.

---

> ### Author Response · Authors · 2023-11-19
> **Response to the comments comments by Reviewer eLXU**
>
> *This is a benchmark paper on simulation on a regular grid. I am not sure if I missed this, the discussion on performance/adaptability of different models on irregular grids is not included. I found the overall analysis and insights comprehensive but also a little bit simple with missing key discussion on a simulation where irregular grids are required.*
>
> **Response:** We thank the reviewer for this invaluable suggestion. We have now included benchmarking of all the models on two additional datasets, namely, Elasticity and Airfoil. Both datasets have general geometries that test the capabilities of models on irregular grid data. To demonstrate this, we have now added modifications to FNO and LSM models, namely, Geo-FNO and Geo-LSM, respectively, to work on irregular grids. See Table 2 and section 5.1 for more details.
>
> *I am curious about more details on how hyperparameters are set for different model training across dataset.*
>
> **Response:** The hyperparameters were chosen based on grid-search. We have now included further details about training methodology, hyperparameter tuning, and sensitivity in the Appendix (see Appendix section F).
>
> **Appeal to the reviewer:** With the additional experiments, results, and explanations, we now hope the reviewer finds the manuscript suitable for publication. Accordingly, we request you to raise the score for the manuscript. Please do let us know if there are any further queries.

---

> > ### Author Response · Authors · 2023-11-22
> > **Keenly awaiting feedback!**
> >
> > Dear Reviewer eLXu,
> >
> > As the author-reviewer discussion phase is approaching its conclusion in just a few hours, we are reaching out to inquire if any remaining concerns or points require clarification. Specifically, to address the comments raised, we have performed additional experiments by considering **two new datasets** on irregular grids and considered two new models, namely, **geo-LSM**, and **geo-FNO**. With these, we hope we have addressed all the outstanding concerns of the reviewer. We request you to raise any additional concerns you have. Otherwise, we would really appreciate if you can raise your score for the paper.
> >
> > Your support in this final phase, particularly if you find the revisions satisfactory, would be immensely appreciated.

---

### Author Response · Authors · 2023-11-19
**General comments. Applies to all the reviewers**

We thank the reviewers for the careful evaluation and suggestions. Please find a point-by-point response to all the comments raised by the reviewers below. We have also updated the main manuscript and the appendix to address these comments. The changes made in the main manuscript are highlighted in blue color. The major changes made in the manuscript are listed below.

1. **Experiments on irregular grid:** Additional experiments on irregular grids are now performed on two new datasets namely, elasticity and airfoil **(see Tab. 2 and Sec. 5.1)**. Moreover, two additional models, namely, Geo-FNO, and Geo-LSM are implemented to evaluate on irrgeular grids.
2. **Additional experiment on super-resolution:** We have now added additional experiments on superresolution on the strain dataset **(see Tab. 5)**.
3. **Latent spectral model:** An additional model namely, Latent spectral model (LSM)[1] is now included for all the datasets.
[1] Solving High-Dimensional PDEs with Latent Spectral Models ICML 23
4. **Additional ReadMe for usability**: To ensure that the package is easy to use, we have added extensive additional information including a detailed ReadMe file so that the readers directly use all the models on the datasets included. Further, all the hyperparameters and datasplit is included for transparency.

---

### Author Response · Authors · 2023-11-21
**Awaiting post-rebuttal feedback**

Dear Reviewers,

Thank you once again for all of your constructive comments, which have helped us significantly improve the paper! As detailed below, we have performed several additional experiments and analyses to address the comments and concerns raised by the reviewers.

Since we are into the last days of the discussion phase, we are eagerly looking forward to your post-rebuttal responses.

Please do let us know if there are any additional clarifications or experiments that we can offer. We would love to discuss more if any concern still remains. Otherwise, we would appreciate it if you could support the paper by increasing the score.

Thank you!